# HOGDA: Boosting Semi-supervised Graph Domain Adaptation via High-Order Structure-Guided Adaptive Feature Alignment

## ABSTRACT

Semi-supervised graph domain adaptation, as a subfield of graph transfer learning, seeks to precisely annotate unlabeled target graph nodes by leveraging transferable features acquired from the limited labeled source nodes. However, most existing studies often directly utilize GCNs-based feature extractors to capture domain-invariant node features, while neglecting the issue that GCNs are insufficient in collecting complex structure information in graph. Considering the importance of graph structure information in encoding the complex relationship among nodes and edges, this paper aims to utilize such powerful information to assist graph transfer learning. To achieve this goal, we develop an novel framework called HOGDA. Concretely, HOGDA introduces a high-order structure information mixing module to effectively assist the feature extractor in capturing transferable node features. Moreover, to achieve fine-grained feature distributions alignment, the AWDA strategy is proposed to dynamically adjust the node weight during adversarial domain adaptation process, effectively boosting the model's transfer ability. Furthermore, to mitigate the overfitting phenomenon caused by limited source labeled nodes, we also design a TNC strategy to guide the unlabeled nodes to achieve discriminative clustering. Extensive experimental results show that our HOGDA outperforms the state-of-the-art methods on various transfer tasks.

## KEYWORDS

Graph Transfer Learning, Adversarial Domain Adaptation, High-Order Moment, Node Clustering

## 1 INTRODUCTION

In multimodal applications, graphs are often used to model the correlation between different modal data. Among them, graph node classification techniques play a crucial role in analyzing nodes from different modalities. However, due to the distribution shift problem, well-trained models suffer severe performance degradation when applied straight to new domains, limiting the large-scale implementation of deep models in actual applications. Graph transfer learning (GTL) [6, 24] has been proposed as a paradigm to address such a problem by transferring some invariant features from a labeled source graph to an unlabeled target graph, greatly improving the model's generalization ability on the target graph.

*ACM MM, 2024, Melbourne, Australia*
© 2024 Copyright held by the owner/author(s). Publication rights licensed to ACM.
ACM ISBN 978-x-xxxx-xxxx-x/YY/MM
https://doi.org/10.1145/nnnnnnn.nnnnnnn

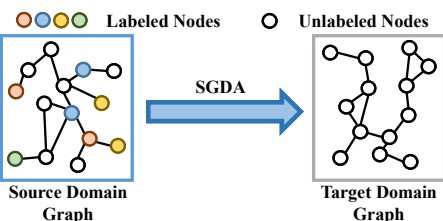

**Figure 1: Illustration of the semi-supervised graph domain adaptation (SGDA) task.**

Most existing studies [7, 18, 28] on GTL tends to focus on unsupervised domain adaptation, assuming that all nodes in the source graph are labeled, while overlooking the fact that this ideal scenario is not common in real-world applications, as annotating the entire source graph is a time-consuming task, especially for large-scale graphs. Therefore, in this paper, we focus on a more practical application scenario known as semi-supervised graph domain adaptation (SGDA) [24], where the source graph contains only a limited number of labeled nodes, as shown in Figure 1. The most crucial challenge for SGDA is to effectively leverage the transferable features learned from the label-scarce source graph to precisely annotate nodes in the target graph.

Unlike images and time series data, graph data usually contains rich structure information that encodes complex relationships among nodes and edges. Most existing GTL models [6, 24, 36] usually adopt graph convolutional network (GCN)-based feature extractors to learn domain-invariant node features. However, recent studies [5, 21, 40] have demonstrated that GCNs are insufficient in capturing the sophisticated structure information in graph, which may potentially affect the transfer of domain-invariant knowledge and consequently limit the model's generalization capability.

To address this problem, inspired by the effectiveness of high-order moment features in characterizing the data structure [9], we propose an novel SGDA framework named **HOGDA** that employs a High-order Structure Information Mixing (**HSIM**) module to effectively capture graph structure information. In order to further explain our motivation, we plot a point cloud (sampled from three Gaussian distributions) and visualize the moment features of different orders in Figure 2. As can be seen, the structure of the point cloud can be captured more accurately by using high-order moment features than by using low-order features. To this end, HSIM module seek to employ multi-view structure information to assist the feature extractor in extracting more discriminative domain-invariant node features.

Recent studies [4, 19, 42] on TL have demonstrated that different samples have different levels of transferability. However, many existing GTL methods usually assign equal weight to different nodes during adversarial domain adaptation process, ignoring the fact that hard-to-transfer nodes may harm the learning of transferable node

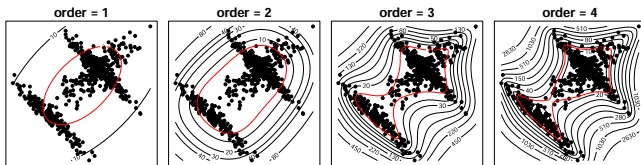

**Figure 2: 240 data points sampled from three Gaussian distributions. And the level sets denotes the moment features with different orders. In comparison to low-order features, high-order moment features can more accurately capture the underlying structure of the point cloud.**

features and even cause negative transfer. To remedy this issue, we propose a Adaptive Weighted Domain Alignment (**AWDA**) strategy. Specifically, AWDA adaptively estimates the node transferability by jointly leveraging entropy information from the classifier and discriminator, and dynamically adjusts the weight of node based on its transferability. By prioritizing easy-to-transfer nodes with higher weights during domain alignment process, the model can learn more domain-invariant features, thereby significantly improving its transfer ability on the target domain.

Furthermore, in the SGDA scenarios, due to the label scarcity of source graph, well-trained model on only a few labeled source nodes is prone to overfitting. Consequently, it may make ambiguous or even incorrect predictions for certain target graph nodes located near the decision boundaries or far from their corresponding class centers. To alleviate this overfitting phenomenon, we devise a Trust-aware Node Clustering (**TNC**) strategy to enhance the model's generalization performance. Specially, TNC aims to guide the discriminative clustering of unlabeled nodes by minimizing the discrepancy between the current cluster assignment distribution and the ideal cluster distribution, effectively promoting the alignment of category distributions across domains.

The following are the primary contributions of this paper:
**(1)** A novel HSIM module is devised to assist the feature extractor in capturing the graph structure information.
**(2)** A node re-weighted adversarial adaption strategy named AWDA is proposed to facilitate the alignment of feature distributions.
**(3)** To remedy overfitting issue, a simple but effective TNC strategy is introduced to guide the clustering of unlabeled nodes.
**(4)** Experimental results on various transfer tasks demonstrate the superiority of our HOGDA over the state-of-the-art methods.

## 2 RELATED WORKS

**Graph Transfer Learning (GTL).** GTL has attracted substantial attention as a promising solution for effectively alleviating the burden of collecting labeled data for novel tasks. A series of early studies commonly utilize available source labeled nodes to build a pre-train graph model for different but related tasks in the target domain [13, 23, 26]. However, due to the presence of distribution shift, this training paradigm inevitably causes the model to suffer severe performance degradation in the target domain. To solve this problem, recent studies have shifted their focus towards domain adaptation [7, 18, 28]. These works aims to boost the model's generalization ability by transferring some invariant features from a label-rich source domain to a label-scarce target domain. Methods for achieving domain adaptation can be roughly divided into two categories: (1) Extracting transferable features by minimizing the statistical metrics between two domains [10, 29]; (2) Leveraging adversarial training to enforce domain confusion to capture domain-invariant features [6, 24, 28, 36, 41].

**Semi-supervised Learning on Graphs.** Semi-supervised learning on graphs aims to tackles the node classification task by utilizing only a small fraction of labeled nodes. Early works, such as GCN [16], GraphSAGE [12] and GAT [34], typically employ the message passing paradigm to capture discriminative node features [12, 16, 34]. In recent studies, researchers have investigated a variety of techniques, such as adversarial training [14, 38], data augmentation [35], continuous graph [37], and meta-learning [25] to further boost the model's generalization performance.

## 3 METHODOLOGY

### 3.1 Problem Formulation

**Source Domain Graph**: Let $\mathcal{G}^s = (\mathcal{V}^{s,l}, \mathcal{V}^{s,u}, A^s, X^s, Y^{s,l})$ be the source graph, where $\mathcal{V}^{s,l}$ denotes the labeled node set, and $\mathcal{V}^{s,u}$ denotes the remaining unlabeled node set in $\mathcal{G}_s$. The adjacency matrix $A^s \in \mathbb{R}^{N^s \times N^s}$ represents the connectivity of nodes in $\mathcal{G}^s$, where $N^s = |\mathcal{V}^{s,l}| + |\mathcal{V}^{s,u}|$ denotes the total number of nodes. If there exists an edge between nodes $n_i$ and $n_j$, the corresponding element $A_{ij}^s$ is assigned a value of 1; otherwise, it is set to 0. $Y^{s,l} \in \mathbb{R}^{|\mathcal{V}^{s,l}| \times C}$ indicates the label matrix of $\mathcal{V}^{s,l}$, where $C$ is the number of node classes. If a node $n_i^s \in \mathcal{V}^{s,l}$ belongs to the $c$-th class, $y_{i,c}^s = 1$; otherwise, $y_{i,c}^s = 0$. $X^s \in \mathbb{R}^{N^s \times e}$ represents an attribute matrix, where $e$ is the dimension of node attributes. In the SGDA setting, $|\mathcal{V}^{s,l}|$ is much smaller than $|\mathcal{V}^{s,u}|$.

**Target Domain Graph**: The target graph, denoted as $\mathcal{G}^t = (\mathcal{V}^t, A^t, X^t)$, is a completely unlabeled graph with an unlabeled node set $\mathcal{V}^t$. Similarly, the adjacency matrix $A^t \in \mathbb{R}^{N^t \times N^t}$ indicates the connections between nodes in $\mathcal{G}_t$, and the node attribute matrix $X^t \in \mathbb{R}^{N^t \times e}$ stores the attribute information for each target node. Here, $N^t = |\mathcal{V}^t|$ denotes the number of nodes in $\mathcal{G}_t$.

**Semi-Supervised Graph Domain Adaptation (SGDA)**: Given a partially labeled source graph $\mathcal{G}^s$ and an unlabeled target graph $\mathcal{G}^t$, the key challenge in SGDA is how to accurately annotate target graph nodes by leveraging the transferable knowledge learned from the limited source labeled nodes.

### 3.2 Network Architecture

The architecture of our HOGDA model is composed of four components: a GCN-based feature extractor $\mathcal{F}$, a high-order structure information mixing (**HSIM**) module $\mathcal{H}$, a domain discriminator $\mathcal{D}$, and a node classifier $C$, as shown in Figure 3.

For brevity, we omit the domain-specific notation to describe the data flow through our model. Mathematically, given an input graph $\mathcal{G} = (\mathcal{V}, A, X)$, the node features extracted by $\mathcal{F}$ is denoted as $Z = \mathcal{F}(\mathcal{G}) \in \mathbb{R}^{|\mathcal{V}| \times e}$, and it is subsequently fed into the HSIM module to capture the corresponding high-order structure features $H = \mathcal{H}(Z) \in \mathbb{R}^{|\mathcal{V}| \times e}$, where $e$ is the feature dimension and $|\mathcal{V}|$ denotes the number of nodes in $\mathcal{G}$. Then, the node features $Z$ are concatenated with their corresponding structure features $H$ and

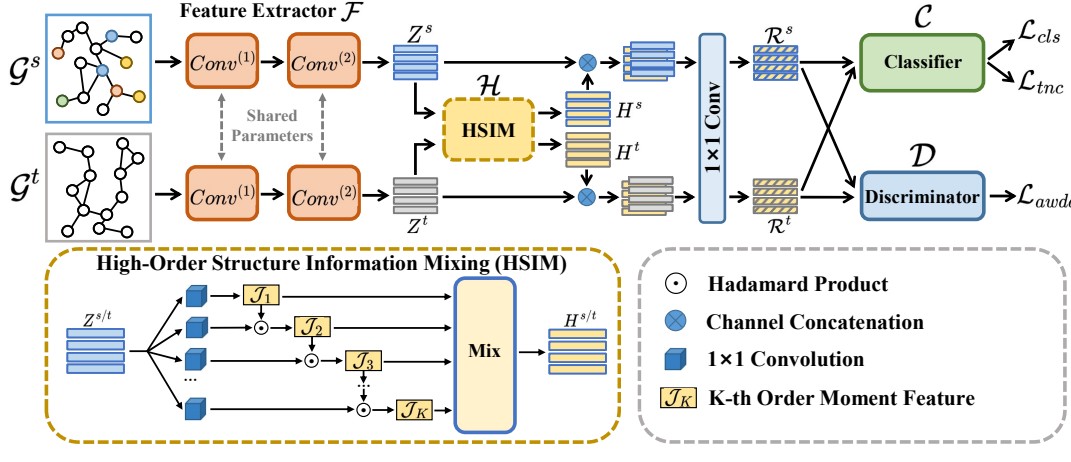

**Figure 3: Global overview of the proposed HOGDA model.**

passed through a learnable $1 \times 1$ convolution filter $\mathcal{W}^{1\times1}$ to obtain the mixed node features $\mathcal{R} \in \mathbb{R}^{|\mathcal{V}|\times e}$. These mixed features are then passed into the classifier $\mathcal{C}$ for the final classification prediction $\mathcal{C}(\mathcal{R}) \in \mathbb{R}^{|\mathcal{V}|\times C}$. The domain discriminator $\mathcal{D}$ is trained to distinguish between the source and target domains, while the feature extractor $\mathcal{F}$ is optimized to confuse $\mathcal{D}$ in order to capture domain-invariant node features $Z$.

To better model the adjacency relationships among nodes in graph $\mathcal{G}$, we calculate the positive point-wise mutual information (PPIM) between nodes following [24, 36]. Concretely, for a given graph $\mathcal{G} = (\mathcal{V}, A, X)$, we employ random walk to sample a set of paths on $A$ and construct a frequency matrix $\Psi$. Here, each entry $\Psi_{ij}$ represents the occurrence count of node $n_j$ within a predefined window in the context of node $n_i$. Then the PPIM matrix $\mathbb{P}$ is computed as:

$$\mathbb{P}_{ij} = \frac{\Psi_{ij}}{\sum_{i,j}\Psi_{ij}}, \quad \mathbb{P}_{i,*} = \frac{\sum_j \Psi_{ij}}{\sum_{i,j}\Psi_{ij}}, \quad \mathbb{P}_{*,j} = \frac{\sum_i \Psi_{ij}}{\sum_{i,j}\Psi_{ij}},$$

$$P_{ij} = \max\{\log(\frac{\mathbb{P}_{ij}}{\mathbb{P}_{i,*} \times \mathbb{P}_{*,j}}), 0\}, \quad (1)$$

where $P_{ij}$ denotes the positive mutual information between nodes $n_i$ and $n_j$, which quantifies the topological proximity between nodes. A higher value of $P_{ij}$ indicates a strong connection between $n_i$ and $n_j$, while a value of $P_{ij} = 0$ indicates the absence of such a connection. Then, the output of the $l$-th GCN layer $Conv^{(l)}(\cdot)$ is denoted as:

$$Z^{(l)} = Conv^{(l)}(P, Z^{(l-1)}) = \sigma(D^{-\frac{1}{2}}\widetilde{P}D^{-\frac{1}{2}}Z^{(l-1)}W^{(l)}), \quad (2)$$

where $\sigma(\cdot)$ is an activation function, and $D$ is the diagonal degree matrix of $P$ (i.e., $D_{ii} = \sum_j \widetilde{P}_{ij}$). Moreover, $\widetilde{P} = P + I$, where $I$ is an identity matrix. $W^{(l)}$ denotes the learnable parameters of the $l$-th layer, and $Z^{(0)} = X$. Note that the feature extractor $\mathcal{F}$ consists of sequentially stacked $L$ layers of GCN $Conv^{(l)}(l = 1, 2, \cdots, L)$.

### 3.3 High-order Structure Information Mixing

As mentioned in Section 1, in contrast to images and time series data, graph data (*e.g.*, social network and academic network) typically encompasses abundant structure information that encodes intricate relationships among nodes and edges. However, most existing GTL models [6, 24, 36] typically employ GCN-based feature extractors to learn domain-invariant node features, neglecting the issue that GCNs are insufficient in collecting complex structure information in graph [5, 21, 40]. This limitation may potentially affect the transfer of domain-invariant knowledge and consequently limit the model's generalization capability.

To address this issue, motivated by the effectiveness of high-order moment information in capturing the structure of data (as shown in Figure 2), we introduce a high-order structure information mixing (**HSIM**) module to capture the graph structure information from multiple views. Specially, HSIM module seeks to leverage these multi-view structure information to assist the feature extractor $\mathcal{F}$ in learning more discriminative transferable node features.

Let $\mathcal{J}_k(Z)$ denote a $k$-th order moment feature, where $Z \in \mathbb{R}^{|\mathcal{V}|\times e}$ is the deep node features extracted by $\mathcal{F}$. Most current works commonly adopt Kronecker product-based approach to calculate high-order moment features of data [3, 11, 33]. However, considering the numerous nodes contained in the graph (e.g., *ACMv9* has over 9000 nodes), directly calculating the Kronecker product of node features is computationally expensive and time-consuming, making it unsuitable for GTL tasks.

To solve this problem, we utilize the Random Maclaurin factorization scheme [15] to achieve the efficient computation of high-order moment. Concretely, as depicted in Figure 3, we employ multiple $1 \times 1$ convolution kernels as several random projectors to estimate the $k$-th order moment features:

$$\mathcal{J}_k(Z) \approx \mathcal{B}_1(Z) \odot \mathcal{B}_2(Z) \odot \cdots \odot \mathcal{B}_k(Z) \in \mathbb{R}^{|\mathcal{V}|\times e} \quad (3)$$

where $\odot$ is the Hadamard product, and $\mathcal{B}_1, \mathcal{B}_2, \cdots, \mathcal{B}_k$ refer to $k$ randomly initialized $1 \times 1$ convolution kernels. However, since the estimators generated by Random Maclaurin scheme are independently of the analyzed distributions [27], which may result in the estimated high-order moment features containing some non-informative high-order components (*i.e.*, noisy components).

To eliminate the impact of these noisy components, we choose to learn the parameters of these projectors (*i.e.*, the $1 \times 1$ convolution kernels) directly from the input data. Note that because the calculation of high-order moments involves a large number of Hadamard product operations, it may cause the estimated high-order features degrade into low-order features. To prevent this degradation phenomenon, we employ a recursive mechanism to progressively approximate the high-order moment features:

$$\mathcal{J}_k(Z) = \mathcal{J}_{k-1}(Z) \odot \mathcal{B}_k(Z). \tag{4}$$

Because different order statistics can capture the graph structure information from different views, we mix various order moment features to more comprehensively capture the graph structure features:

$$H = \mathcal{H}(Z) = (\sum_{k=1}^{K} \mathcal{J}_k(Z)) \in \mathbb{R}^{|\mathcal{V}| \times e} \tag{5}$$

To leverage such powerful structure information to assist the feature extractor $\mathcal{F}$ in capturing discriminative transferable features, we concatenate the multi-view structure features $H$ with the deep node features $Z$ extract by $\mathcal{F}$ to obtain the augmented node features $[H; Z] \in \mathbb{R}^{|\mathcal{V}| \times e \times 2}$. After that, to adaptively combine the advantages of the node features and the structure features, a learnable $1 \times 1$ convolution filter $\mathcal{W}^{1 \times 1}$ is utilized to integrate these features:

$$\mathcal{R} = \mathcal{W}^{1 \times 1}([H; Z]) \in \mathbb{R}^{|\mathcal{V}| \times e}. \tag{6}$$

Then the mixed node features $\mathcal{R}$ will be fed into the classifier $C$ for the final prediction. Given the source labeled node set $\mathcal{V}^{s,l}$, the supervised classification loss on the source graph $\mathcal{G}^s$ can be formulated as:

$$\mathcal{L}_{cls} = \frac{1}{|\mathcal{V}^{s,l}|} \sum_{n_i^s \in \mathcal{V}^{s,l}} \mathcal{L}_{ce}(C(r_i^s), y_i^s). \tag{7}$$

where $\mathcal{L}_{ce}$ is the standard cross-entropy loss, and $r_i^s \in \mathbb{R}^e$ denotes the $i$-th node feature in node features matrix $\mathcal{R}^s$.

It is worth mentioning that the integration of graph structure information is beneficial in guiding unlabeled nodes to achieve discriminative clustering, thus facilitating the learning of fine-grained domain-invariant features, as verified in Figure 4.

## 3.4 Adaptive Weighted Domain Alignment

Transferability denotes the ability of sample feature to bridge the discrepancy across domains. It has recently been demonstrated that, in real scenarios, different samples have different levels of transferability [4, 19, 42]. Specially, some samples contain more transferable features, which we term easy-to-transfer samples, and generally have higher transferability. In contrast, hard-to-transfer samples are difficult for the model to capture their transferable features and generally exhibit lower transferability.

Adversarial training has been widely adopted by existing GTL models to extract domain-invariant node features. However, most existing methods [6, 24, 28, 41] typically assign equal weight to different nodes during adversarial domain adaption, ignoring the fact that hard-to-transfer nodes may harm the learning of domain-invariant features and even lead to negative transfer.

To address this issue, motivated by curriculum learning [2], we propose an Adaptive Weighted Domain Alignment (**AWDA**) strategy, which dynamically adjusts the weight of each node based on its transferability. Specially, **(1) During the early training stage**, the model aims to roughly align the marginal feature distributions through adversarial training. Therefore, at this stage, node transferability should primarily be estimated by the discriminator $\mathcal{D}$. The entropy $\mathcal{E}_D$ of discriminator output can be regarded as a good indicator to measure node transferability. At such stage, easy-to-transfer nodes generally have significantly higher entropy $\mathcal{E}_D$ than hard-to-transfer nodes. **(2) During the mid and late training stages**, the model primarily focuses on aligning the category distributions across domains. In such case, node transferability should mainly be estimated based on the entropy $\mathcal{E}_C$ of the classifier output. Concretely, at these stages, easy-to-transfer samples generally have relatively certain classification predictions (*i.e.*, low entropy $\mathcal{E}_C$) since they are close to the corresponding class centers. Hard-to-transfer samples scattered near the decision boundaries, typically have uncertain predictions (*i.e.*, high entropy $\mathcal{E}_C$) and are prone to misclassification.

Based on the above analysis, we design a mixed entropy-aware weighted mechanism $w(n_i)$ that combines both the classifier and the discriminator information to adaptively estimate the transferability for each node $n_i$:

$$w(n_i) = 1 + e^{-[(2-d_{\mathcal{A}})\mathcal{E}_C - d_{\mathcal{A}}\mathcal{E}_D]}, \tag{8}$$

where $d_{\mathcal{A}}$ denotes the $\mathcal{A}$-distance [1] $d_{\mathcal{A}} = 2(1 - \epsilon(f))$, which is used to measure distribution discrepancy across domains, where $\epsilon(f)$ is the test error of a binary kernel SVM classifier $f$ trained to distinguish the source and target nodes.

Concretely, **(1) during the early training stage**, the source and target domains exhibit significant discrepancy in the deep feature space, and the binary classifier $f$ can almost perfectly distinguish between them (*i.e.*, $\epsilon(f) \to 0$ and $d_{\mathcal{A}} \to 2$). In this way, node transferability is primarily determined by the discriminator $\mathcal{D}$, *i.e.*, $w(n_i) \approx 1 + e^{d_{\mathcal{A}}\mathcal{E}_D}$. **(2) As training progresses**, when the marginal distributions of two domains almost coincide, the binary classifier $h$ cannot distinguish between them, and thus $\epsilon(f) \to 0.5$ and $d_{\mathcal{A}} \to 0$. In such case, node transferability is mainly estimated by the classifier $C$, *i.e.*, $w(n_i) \approx 1 + e^{-(2-d_{\mathcal{A}})\mathcal{E}_C}$.

Notably, easy-to-transfer nodes usually contain more transferable features, while hard-to-transfer nodes tend to have fewer transferable features. To facilitate the fine-grained alignment of feature distributions and accelerate the learning of domain-invariant features, we encourage the model to pay more attention to easy-to-transfer nodes during adversarial domain adaptation process. Therefore, the node transferability weighted adversarial training loss $\mathcal{L}_{awda}$ can be defined as:

$$\mathcal{L}_{awda} = \frac{1}{N^s} \sum_{i=1}^{N^s} w(n_i^s) \log[\mathcal{D}(T(q_i^s))] + \frac{1}{N^t} \sum_{j=1}^{N^t} w(n_j^t) \log[1 - \mathcal{D}(T(q_j^t))]. \tag{9}$$

where $q_i = (r_i, C(r_i))$ is the joint variable of mixed node feature $r_i$ and its corresponding classifier prediction $C(r_i)$. $T(\cdot)$ is a multilinear map employed to promote the alignment of multi-modal category distributions as previous study [19].

In this way, the model will focus on the learning of easy-to-transfer samples to roughly align the two domains during the early training stage. As the distribution discrepancy decreases, hard-to-transfer samples get more attention in the adversarial alignment process and are gradually turned into easy-to-transfer ones.

## 3.5 Trust-aware Node Clustering

Due to the limited number of labeled nodes in $\mathcal{G}^s$, the model is likely to encounter overfitting issue if it simply relies on $\mathcal{L}_{cls}$ for optimization, which may severely degrades the model's generalization performance on $\mathcal{G}^t$. Existing studies usually adopt pseudo-labels strategy [24] or conditional entropy term [36] to guide the learning of unlabeled nodes to mitigate this overfitting phenomenon.

However, there is a concern that the pseudo-labels based strategy inevitably introduces noise into the model and minimizing the conditional entropy term may lead to degenerate clustering solutions (i.e., all unlabeled nodes are assigned to the same cluster), which severely affect the alignment of feature distributions.

To alleviate this overfitting issue, we devise a innovative Trust-aware Node Clustering (**TNC**) strategy to enhance the model's robustness. Considering that both the spatial prototype information and the classifier prediction information can estimate the cluster assignment of node from different views during training, TNC aims to adaptively combine these information to guide the discriminative clustering of unlabeled nodes.

Specially, we first utilizes source labeled nodes to approximate the class centers (i.e., prototypes) $\mu_c^s$ of the source domain:

$$\mu_c^s = \frac{1}{M} \sum_{i=1}^{|\mathcal{V}^{s,l}|} r_i^s \cdot \phi(y_i^s, c), \tag{10}$$

where $\phi(y_i^s, c) = 1$ if $y_i^s = c$, otherwise $\phi(y_i^s, c) = 0$. $c \in \{1, 2, \cdots, C\}$ is the class indicator and $M = \sum_{i=1}^{|\mathcal{V}_{s,l}|} \phi(y_i^s, c)$.

On one hand, the spatial prototype information can estimate cluster assignment for each node $n_i$ by measuring the similarity between the node feature $r_i$ and the class center $\mu_c^s$:

$$\mathbb{S}(i, c) = \frac{\exp^{-\gamma(r_i, \mu_c^s)}}{\sum_{c'=1}^{C} \exp^{-\gamma(r_i, \mu_{c'}^s)}}, \tag{11}$$

where $\mathbb{S}(i, c)$ denotes the probability of assigning node $n_i$ to the $c$-th cluster, and $\gamma(r_i, \mu_c^s)$ denotes the similarity between the node features $r_i$ and the $c$-th class centers $\mu_c^s$. In our experiment, we employ the Student's $t$-distribution based kernel strategy [32] as the similarity metric $\gamma(\cdot, \cdot)$, which can be defined as:

$$\gamma(r_i, \mu_c^s) = \frac{\exp((1 + \frac{\|r_i - \mu_c^s\|^2}{\alpha}))^{-\frac{\alpha+1}{2}}}{\sum_{c'=1}^{C} \exp((1 + \frac{\|r_i - \mu_{c'}^s\|^2}{\alpha}))^{-\frac{\alpha+1}{2}}}, \tag{12}$$

where $\alpha$ is the degree of freedom of the Student's $t$-distribution. In this work, $\alpha$ is set to 1 for all experiments.

One the other hand, the classifier $C$ can also predict the cluster assignment for each input node $n_i$:

$$\mathbb{W}(i, c) = Pro(y_i = c | C(r_i)), \tag{13}$$

where $\mathbb{W}(i, c)$ denotes the probability of node $n_i$ belonging to class $c$.

Although both $\mathbb{S}(i, c)$ and $\mathbb{W}(i, k)$ can measure the probability of assigning node $n_i$ to cluster $c$, their confidence changes dynamically during training. To this end, we introduce a trustworthy weighted mechanism to dynamically adjust their importance:

$$\Omega(i, c) = \frac{d_{\mathcal{A}}\mathbb{S}(i, c) + (2 - d_{\mathcal{A}})\mathbb{W}(i, c)}{\sum_{c'=1}^{C}(d_{\mathcal{A}}\mathbb{S}(i, c') + (2 - d_{\mathcal{A}})\mathbb{M}(i, c'))}, \tag{14}$$

where $\Omega(i, c)$ can estimate the probability of node $n_i$ belonging to the $c$-th cluster in a more robust manner, and $d_{\mathcal{A}}$ denotes the $\mathcal{A}$-distance between two domains. Specially, **during the early training stage**, as the classifier $C$ is learned from scratch, $\mathbb{S}(i, c)$ is much more reliable than $\mathbb{W}(i, k)$. In such case, the clustering assignment of node is primarily estimated by the spatial prototype information(i.e., $\epsilon(f) \to 0$ and $d_{\mathcal{A}} \to 2$). **During the mid and late training stages**, as the category distributions are gradually aligned (i.e., $\epsilon(f) \to 0.5$ and $d_{\mathcal{A}} \to 0$), the classifier $C$ can provide more precise estimations for cluster assignment. In such case, the term $\mathbb{W}(i, c)$ in $\Omega(i, c)$ becomes the main contributor.

Notably, the ideal clustering should satisfy these two conditions: **i)** The clustering assignment $\Omega$ of each node should be sufficiently certain; **ii)** Each cluster should contain some nodes (i.e., degenerate solutions will not occur). Here, $\Omega \in \mathbb{R}^{|\mathcal{V}|\times C}$ is the clustering assignment matrix of all nodes, which can be viewed as a distribution.

To achieve this goal, we define a ideal clustering distribution $\Phi \in \mathbb{R}^{|\mathcal{V}|\times C}$ and encourage the current cluster assignment distribution $\Omega$ to approach the ideal distribution $\Phi$ by using the following objective function $\mathcal{I}$:

$$\begin{aligned}
\mathcal{I} &= KL(\Phi||\Omega) + KL(\rho||u) \\
&= \left[\frac{1}{|\mathcal{V}|} \sum_{i=1}^{|\mathcal{V}|} \sum_{c=1}^{C} \Phi(i, c) \log \frac{\Phi(i, c)}{\Omega(i, c)}\right] + \left[\frac{1}{|\mathcal{V}|} \sum_{c=1}^{C} \rho_c \log \frac{\rho_c}{u_c}\right] \\
&= \frac{1}{|\mathcal{V}|} \sum_{i=1}^{|\mathcal{V}|} \sum_{c=1}^{C} \Phi(i, c) \log \frac{\Phi(i, c)}{\Omega(i, c)} + \Phi(i, c) \log \frac{\rho_c}{u_c},
\end{aligned} \tag{15}$$

where $KL$ represents the Kullback-Leiber divergence, $u$ is the uniform prior, and $\rho_c$ denotes the soft frequency of cluster assignments in the ideal distribution $\Phi$:

$$\rho_c = \frac{1}{|\mathcal{V}|} \sum_i \Phi(i, c). \tag{16}$$

Specially, in Eq. 15, the first $KL$ term denotes the discrepancy between the current cluster assignment $\Omega$ and the ideal target $\Phi$. The second $KL$ term is used to promote balanced cluster assignments in order to avoid degenerate solutions.

To estimate $\Phi$, we employ iterative learning mechanism to optimize this objective function $\mathcal{I}$. Concretely, in each training iteration, assuming the network parameters $\theta$ are fixed, we can infer the variable $\Phi$ by solving the following optimization problem:

$$\min_{\Phi} \frac{1}{|\mathcal{V}|} \sum_{i=1}^{|\mathcal{V}|} \sum_{c=1}^{C} \Phi(i, c) \log \frac{\Phi(i, c)}{\Omega(i, c)} + \Phi(i, c) \log \frac{\rho_c}{u_c}, \tag{17}$$
$$s.t. \sum_c \Phi(i, c) = 1.$$

As this optimization problem can be effectively solved by utilizing several gradient-based algorithms [22] (such as Nesterov optimal and projected gradient descent methods), we can calculate the partial derivative of function $\mathcal{I}$ with respect to variable $\Phi$ as:

$$\frac{\partial \mathcal{I}}{\partial \Phi(i,c)} \propto \log\left(\frac{\Phi(i,c)\rho_c}{\Omega(i,c)}\right) + \frac{\Phi(i,c)}{\sum\limits_{i'=1}^{|\mathcal{V}|} \Phi(i',c)} + 1, \qquad (18)$$

Since the number of nodes $|\mathcal{V}|$ is typically very large, we can approximate the gradient in Eq. 18 by neglecting the second term. In this way, we obtain an approximate closed-form solution for $\Phi$ by setting the gradient to zero:

$$\Phi(i,c)^* = \frac{\Omega(i,c)/(\sum_{i'} \Omega(i',c))^{\frac{1}{2}}}{\sum\limits_{c'} \Omega(i,c')/(\sum_{i'} \Omega(i',c'))^{\frac{1}{2}}}. \qquad (19)$$

In TNC strategy, both source and target domain nodes $\mathcal{V}^{s,l} \cup \mathcal{V}^{s,u} \cup \mathcal{V}^t$ are used to compute the clustering assignment matrix $\Omega$. Therefore, the loss function $\mathcal{L}_{tnc}$ of TNC strategy can be defined as:

$$\mathcal{L}_{tnc} = KL(\Phi^*||\Omega) + KL(\rho^*||u). \qquad (20)$$

The reason we employ source labeled nodes $\mathcal{V}^{s,l}$ in TNC strategy is because they can effectively guide the discriminative clustering of unlabeled nodes towards the desired direction. Notably, our TNC strategy does not involve any pseudo-labels, which not only enhances the model's robustness, but also promotes the precise alignment of category distributions (see Figure 5 for further analysis).

### 3.6 Model Optimization

To sum up, the total loss function of HOGDA can be formulated as:

$$\min_{\mathcal{F},\mathcal{C},\mathcal{H},\mathcal{W}^{1\times1}} \max_{\mathcal{D}} \mathcal{L}_{cls} + \eta\mathcal{L}_{awda} + \beta\mathcal{L}_{tnc} \qquad (21)$$

where hyper-parameters $\eta$ and $\beta$ are used to balance the contributions of the corresponding term.

## 4 EXPERIMENTS

### 4.1 Setup

**Datasets.** Our experiments encompass three real-world graphs[31]: *ACMv9* (**A**), *Citationv1* (**C**), and *DBLPv7* (**D**). In these graphs, every node corresponds to a paper, and the attribute of each paper is a sparse bag-of-words vector derived from its title. The edges in these graphs depict citation relationships among the papers. Considering that these graphs contain diverse sets of node attributes, we merge their attribute sets and resize the attribute dimension to 6775 following [24]. Each node is assigned a 5-class label, determined by its relevant research areas, including *Artificial Intelligence*, *Computer Vision*, *Database*, *Information Security*, and *Networking*. Six typical cross-domain tasks will be carried out in our experiments: **A→C**, **A→D**, **C→A**, **C→D**, **D→A** and **D→C**. Further settings and implementation can be found in the **Supplementary Materials**.
**Compared Methods.** We mainly compare our method with several SOTA **(1)** graph semi-supervised learning methods and **(2)** graph domain adaptation methods following the pioneering work [24]: **(1)** GCN [16], **GSAGE** [12], **GAT** [34], **GIN** [39], **(2)** DANN [8],

CDAN [20], **UDA-GCN** [36], **AdaGCN** [6] and **SGDA** [24]. Note that **DANN**$_{GCN}$ and **CDAN**$_{GCN}$ are two variants that replace the MLP-based encoders with GCN-based feature extractors.
**Evaluation Metrics.** Following previous works [24, 28], we employ **Micro-F1** and **Macro-F1** as evaluation metrics. We repeat each experiment 5 times and record the average accuracy along with standard deviation. Additionally, to address the impact of randomness, we sample different label sets for each experiment.

### 4.2 Results and Discussion

To demonstrate the superiority of our HOGDA, we follow [24] to evaluate its performance in the challenging scenario, where only **5%** of the nodes in the source graph are labeled. Table 1 lists the classification results of different methods on the target graph.

We can observe that our model obtains the overall best results on all transfer tasks. Concretely, HOGDA significantly outperforms the SOTA competitor SGDA [24] by +8.1% and +10.3% on "Micro-F1" and "Macro-F1" respectively for the **C→A** task, indicating the advantage in extracting discriminative transferable features. Furthermore, HOGDA achieves substantial performance gains in some hard transfer scenarios, such as **D→A** and **C→A**, where the size of target domain is larger than source domain, implying the robustness of our model in face of some small-scale dataset scenarios. Notably, we find that most methods, especially adversarial training-based methods, perform poorly because they can only roughly align the marginal distributions and cannot effectively utilize unlabeled nodes. In contrast, our method addresses these limitations. Additionally, the results with a smaller fluctuation range imply the stability of our model, which further confirms the importance of adaptively guiding domain alignment and node clustering.

### 4.3 Ablation Study and Analysis

Due to page size limitation, more experiments and analysis are given in the **Supplementary Materials**.
**1) Ablation Study:** To analyze the contribution of each component in our model, we compare HOGDA and its 7 variants on different transfer tasks. Table 2 describes the variants of HOGDA, and Table 1 shows the results of ablation study.
**Contribution of Each Component:** The results in Table 1 reflect the following observations: **(1)** Due to the label scarcity in source domain, HOGDA-S (baseline) inevitably suffers from overfitting issues, resulting in poor generalization performance on all tasks. **(2)** Variants HOGDA-H, HOGDA-A and HOGDA-T greatly outperform HOGDA-S on all tasks, implying that incorporating high-order structure information, conducting node weighted domain alignment, and guiding discriminative clustering of unlabeled nodes all effectively promote the learning of domain-invariant node features.
**Correlation of Our Strategies:** The results in Table 1 show that combining different strategies can significantly boost the model's transfer ability, suggesting a distinct complementary relationship among the HSIM, AWDA, and TNC strategies.
**2) Node Features Visualization:** To showcase the superior transfer ability of our model, we utilize t-SNE [32] to visualize the node features on task **A→C** under the same 5% label rate setting, as depicted in Figure 4. As for the SOTA method SGDA, the category distributions are not well aligned and the decision boundary is not

**Table 1: Transfer performance (%) on six tasks with a source graph label rate of 5% for semi-supervised graph domain adaptation.**

| Methods | A→C | | A→D | | C→A | | C→D | | D→A | | D→C | |
|---|---|---|---|---|---|---|---|---|---|---|---|---|
| | Micro-F1 | Macro-F1 | Micro-F1 | Macro-F1 | Micro-F1 | Macro-F1 | Micro-F1 | Macro-F1 | Micro-F1 | Macro-F1 | Micro-F1 | Macro-F1 |
| MLP | $41.3_{\pm1.15}$ | $35.8_{\pm0.72}$ | $42.8_{\pm0.88}$ | $36.3_{\pm0.77}$ | $39.4_{\pm0.57}$ | $33.7_{\pm0.58}$ | $43.7_{\pm0.69}$ | $36.7_{\pm0.55}$ | $37.3_{\pm0.32}$ | $30.8_{\pm0.37}$ | $39.4_{\pm0.99}$ | $32.8_{\pm0.99}$ |
| GCN | $54.4_{\pm1.52}$ | $52.0_{\pm1.62}$ | $56.9_{\pm2.33}$ | $53.4_{\pm2.81}$ | $54.1_{\pm1.40}$ | $52.3_{\pm1.98}$ | $58.9_{\pm0.99}$ | $54.5_{\pm1.55}$ | $50.1_{\pm2.14}$ | $48.0_{\pm3.28}$ | $56.0_{\pm1.24}$ | $51.9_{\pm1.49}$ |
| GSAGE | $49.3_{\pm2.18}$ | $46.4_{\pm2.06}$ | $51.8_{\pm1.35}$ | $47.4_{\pm1.62}$ | $46.8_{\pm2.56}$ | $45.0_{\pm2.78}$ | $51.7_{\pm1.95}$ | $48.1_{\pm1.97}$ | $41.7_{\pm2.17}$ | $37.4_{\pm4.59}$ | $45.4_{\pm2.11}$ | $39.3_{\pm3.45}$ |
| GAT | $55.1_{\pm3.22}$ | $50.8_{\pm1.45}$ | $55.3_{\pm2.52}$ | $51.8_{\pm2.60}$ | $50.0_{\pm1.20}$ | $45.6_{\pm2.36}$ | $55.4_{\pm2.73}$ | $49.2_{\pm2.59}$ | $44.8_{\pm2.74}$ | $38.3_{\pm4.84}$ | $50.4_{\pm3.35}$ | $42.0_{\pm4.46}$ |
| GIN | $64.6_{\pm2.47}$ | $56.0_{\pm2.73}$ | $60.0_{\pm2.09}$ | $51.3_{\pm3.99}$ | $57.1_{\pm1.19}$ | $54.4_{\pm2.57}$ | $62.0_{\pm1.05}$ | $56.8_{\pm1.40}$ | $51.9_{\pm2.00}$ | $45.4_{\pm2.16}$ | $60.2_{\pm3.05}$ | $53.0_{\pm2.10}$ |
| DANN | $44.3_{\pm2.03}$ | $39.3_{\pm1.86}$ | $44.0_{\pm1.42}$ | $38.7_{\pm1.47}$ | $41.8_{\pm1.95}$ | $37.6_{\pm1.24}$ | $45.5_{\pm0.71}$ | $39.6_{\pm1.55}$ | $37.8_{\pm3.66}$ | $33.2_{\pm2.23}$ | $41.7_{\pm2.32}$ | $35.6_{\pm2.55}$ |
| CDAN | $44.6_{\pm1.30}$ | $38.6_{\pm1.07}$ | $45.5_{\pm0.85}$ | $38.0_{\pm0.86}$ | $42.4_{\pm0.64}$ | $36.2_{\pm1.17}$ | $46.7_{\pm1.17}$ | $39.2_{\pm0.96}$ | $39.0_{\pm1.08}$ | $32.3_{\pm1.09}$ | $41.7_{\pm1.55}$ | $34.8_{\pm1.56}$ |
| DANN$_{GCN}$ | $63.0_{\pm6.75}$ | $59.6_{\pm6.02}$ | $62.2_{\pm1.90}$ | $57.7_{\pm3.16}$ | $56.7_{\pm0.38}$ | $55.2_{\pm1.03}$ | $65.3_{\pm2.04}$ | $59.0_{\pm2.39}$ | $52.3_{\pm2.59}$ | $48.6_{\pm4.52}$ | $58.1_{\pm2.78}$ | $52.4_{\pm3.81}$ |
| CDAN$_{GCN}$ | $70.3_{\pm0.84}$ | $66.5_{\pm0.66}$ | $65.0_{\pm1.00}$ | $61.3_{\pm0.96}$ | $56.3_{\pm1.78}$ | $53.6_{\pm2.70}$ | $65.2_{\pm2.19}$ | $58.8_{\pm2.38}$ | $53.0_{\pm1.34}$ | $48.7_{\pm3.51}$ | $59.0_{\pm1.52}$ | $53.3_{\pm1.99}$ |
| UDA-GCN | $72.4_{\pm2.75}$ | $65.2_{\pm6.51}$ | $68.0_{\pm6.38}$ | $64.3_{\pm7.12}$ | $62.9_{\pm0.33}$ | $62.2_{\pm1.44}$ | $71.4_{\pm2.56}$ | $67.5_{\pm2.25}$ | $55.8_{\pm3.50}$ | $52.4_{\pm2.68}$ | $65.2_{\pm4.41}$ | $60.7_{\pm6.84}$ |
| AdaGCN | $70.8_{\pm0.95}$ | $68.5_{\pm0.73}$ | $68.2_{\pm3.84}$ | $64.2_{\pm3.91}$ | $61.5_{\pm2.20}$ | $60.4_{\pm3.15}$ | $69.1_{\pm1.96}$ | $65.8_{\pm2.87}$ | $56.1_{\pm1.75}$ | $53.8_{\pm2.95}$ | $64.1_{\pm0.91}$ | $62.8_{\pm1.56}$ |
| SGDA | $75.6_{\pm0.57}$ | $71.4_{\pm0.82}$ | $69.2_{\pm0.73}$ | $64.7_{\pm2.36}$ | $66.3_{\pm0.68}$ | $62.3_{\pm0.96}$ | $72.9_{\pm1.26}$ | $68.9_{\pm1.83}$ | $60.6_{\pm0.86}$ | $56.0_{\pm0.90}$ | $73.2_{\pm0.59}$ | $69.3_{\pm1.01}$ |
| **HOGDA-S** | $55.8_{\pm1.76}$ | $53.6_{\pm1.84}$ | $54.2_{\pm2.11}$ | $44.9_{\pm2.04}$ | $58.2_{\pm1.52}$ | $48.9_{\pm1.94}$ | $57.0_{\pm1.03}$ | $46.3_{\pm1.60}$ | $49.8_{\pm2.33}$ | $41.0_{\pm3.46}$ | $55.9_{\pm1.41}$ | $45.2_{\pm1.72}$ |
| **HOGDA-H** | $73.5_{\pm0.51}$ | $67.1_{\pm0.79}$ | $67.4_{\pm0.82}$ | $63.8_{\pm1.27}$ | $66.0_{\pm0.76}$ | $62.7_{\pm0.85}$ | $67.3_{\pm0.98}$ | $62.2_{\pm1.37}$ | $57.9_{\pm0.81}$ | $55.3_{\pm0.85}$ | $70.6_{\pm0.52}$ | $68.4_{\pm0.94}$ |
| **HOGDA-A** | $76.7_{\pm0.42}$ | $71.2_{\pm0.65}$ | $71.8_{\pm0.59}$ | $68.1_{\pm1.02}$ | $69.1_{\pm0.66}$ | $64.0_{\pm0.94}$ | $72.6_{\pm0.92}$ | $68.7_{\pm1.01}$ | $63.4_{\pm0.76}$ | $59.6_{\pm0.81}$ | $74.8_{\pm0.53}$ | $71.5_{\pm0.90}$ |
| **HOGDA-T** | $74.9_{\pm0.42}$ | $70.8_{\pm0.71}$ | $69.0_{\pm0.49}$ | $64.5_{\pm0.92}$ | $65.8_{\pm0.50}$ | $62.1_{\pm0.83}$ | $70.6_{\pm0.89}$ | $66.6_{\pm1.17}$ | $62.9_{\pm0.65}$ | $57.2_{\pm0.81}$ | $72.7_{\pm0.46}$ | $69.0_{\pm0.73}$ |
| **HOGDA-HA** | $80.7_{\pm0.53}$ | $78.3_{\pm0.59}$ | $74.7_{\pm0.82}$ | $72.2_{\pm1.25}$ | $72.6_{\pm0.64}$ | $71.5_{\pm0.83}$ | $75.2_{\pm1.07}$ | $72.1_{\pm1.26}$ | $65.7_{\pm0.71}$ | $63.3_{\pm0.80}$ | $77.4_{\pm0.59}$ | $73.8_{\pm0.87}$ |
| **HOGDA-HT** | $80.3_{\pm0.62}$ | $77.5_{\pm0.50}$ | $74.2_{\pm0.57}$ | $71.6_{\pm1.09}$ | $72.2_{\pm0.53}$ | $71.1_{\pm0.87}$ | $74.8_{\pm1.05}$ | $71.7_{\pm1.61}$ | $64.9_{\pm0.63}$ | $62.4_{\pm0.75}$ | $78.5_{\pm0.49}$ | $74.0_{\pm0.76}$ |
| **HOGDA-AT** | $81.5_{\pm0.43}$ | $78.7_{\pm0.51}$ | $75.4_{\pm0.50}$ | $73.3_{\pm0.87}$ | $73.1_{\pm0.55}$ | $72.0_{\pm0.89}$ | $75.8_{\pm0.97}$ | $72.4_{\pm1.26}$ | $66.7_{\pm0.51}$ | $64.2_{\pm0.63}$ | $79.1_{\pm0.46}$ | $74.5_{\pm0.82}$ |
| **HOGDA** | $\mathbf{82.4}_{\pm0.36}$ | $\mathbf{79.2}_{\pm0.43}$ | $\mathbf{76.5}_{\pm0.47}$ | $\mathbf{73.8}_{\pm0.82}$ | $\mathbf{74.4}_{\pm0.46}$ | $\mathbf{72.6}_{\pm0.78}$ | $\mathbf{77.1}_{\pm0.93}$ | $\mathbf{72.9}_{\pm1.21}$ | $\mathbf{68.2}_{\pm0.47}$ | $\mathbf{65.1}_{\pm0.56}$ | $\mathbf{80.3}_{\pm0.41}$ | $\mathbf{75.0}_{\pm0.82}$ |

**Table 2: Different variants of HOGDA.**

| Variant | $\mathcal{L}_{cls}$ | HSIM | $\mathcal{L}_{awda}$ | $\mathcal{L}_{tnc}$ |
|---|---|---|---|---|
| HOGDA-S | ✓ | | | |
| HOGDA-H | ✓ | ✓ | | |
| HOGDA-A | ✓ | | ✓ | |
| HOGDA-T | ✓ | | | ✓ |
| HOGDA-HA | ✓ | ✓ | ✓ | |
| HOGDA-HT | ✓ | ✓ | | ✓ |
| HOGDA-AT | ✓ | | ✓ | ✓ |
| **HOGDA** | ✓ | ✓ | ✓ | ✓ |

clear enough. We can see noticeable overlaps between different clusters, which increases the risk of misclassifying hard-to-transfer nodes. In contrast, our HOGDA precisely align the category distributions across domains and achieves exactly 5 clusters with clean decision boundaries, implying that our method can promote transferable node features become more discriminative.

**3) Effect of HSIM:** To demonstrate the effectiveness of HSIM, we conduct in-depth experiments from both quantitative and visual aspects: **(1)** As reported in Table 1, variants HOGDA-HA and HOGDA-HT greatly surpass HOGDA-A and HOGDA-T respectively, implying that the injection of structure information can facilitate the learning of transferable features. **(2)** As illustrated in Figure 4, compared to HOGDA-T, variant HOGDA-HT exhibits better intra-class compactness and inter-class separability in the feature space, indicating that the graph structure information can guide the discriminative clustering of unlabeled nodes, thereby promoting the learning of domain-invariant features.

**4) Effect of TNC:** To showcase the superiority of TNC strategy, we conduct a comparative analysis with existing nodes clustering strategies, including conditional entropy minimization strategy (**ENT**) [36] and the recently proposed posterior scores-based pseudo-labeling strategy (**PSPL**) [24]. We record the trend of Micro-F1 score during training on tasks **A→C** and **A→D**, respectively. The curves in Figure 5 reflect the following observations: **(1)** HOGDA-T exhibits smoother and faster convergence, leading to superior transfer performance, which implies that our TNC strategy can effectively guide the discriminative clustering of unlabeled nodes and promote the fine-grained alignment of category distributions. **(2)** Compared to HOGDA-S (baseline), HOGDA-S + ENT suffers from severe performance degradation, as ENT causes the unlabeled nodes to fall into a degenerate clustering solution. **(3)** PSPL strategy inevitably introduces some pseudo-label noise to the model during training, making it difficult to further improve the model's generalization ability.

**5) Effect of Label Rate:** We investigate the model's performance under different label scarcity settings on two typical transfer tasks: **A→C** and **A→D**. Specially, the source graph is assigned label rates of 1%, 5%, 7%, 9%, and 10% respectively, as depicted in Figure 6. We can find that HOGDA surpasses other competitors by a significant margin, even in the most challenging environment of 1% label rate, which indicating the robustness of HOGDA when facing different challenging transfer scenarios.

**6) Effect of AWDA:** To demonstrate the effectiveness of our AWDA strategy, we compare it with existing domain alignment strategies, including the standard adversarial domain alignment strategy (**AD**) [6], sliced Wasserstein distance-based alignment strategy (**SWD**)[17], class-conditional MMD strategy (**CMMD**) [30], and the recently proposed shifting-guided adversarial domain alignment strategy (**SAD**) [24]. We employ variant HOGDA-S as the baseline

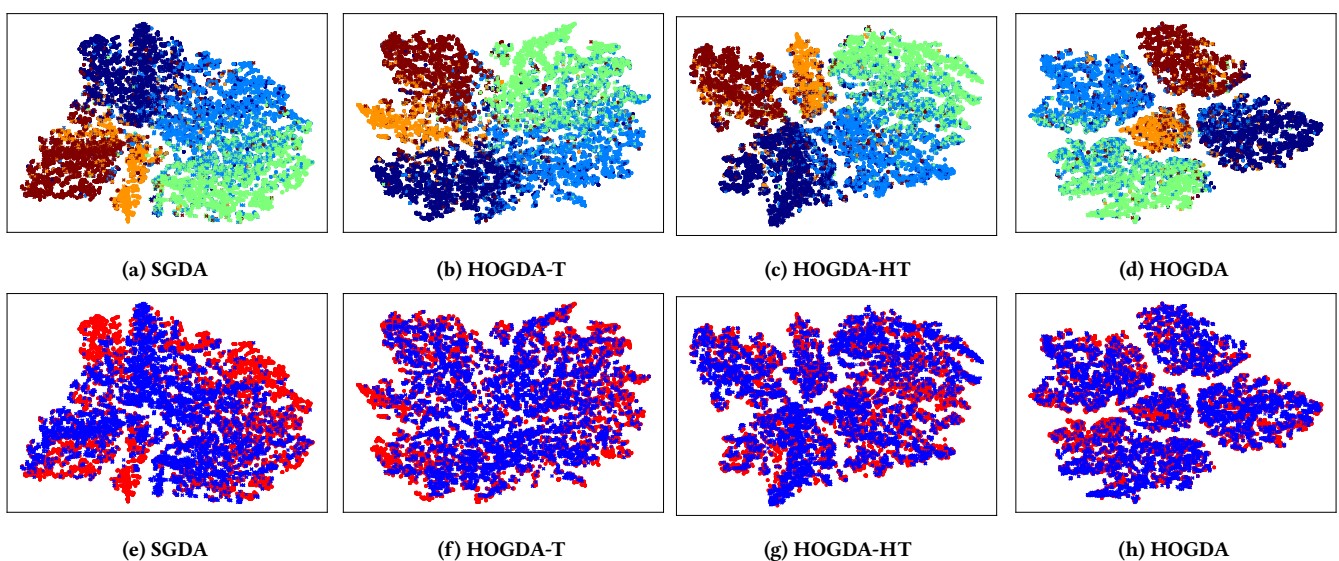

(a) SGDA     (b) HOGDA-T     (c) HOGDA-HT     (d) HOGDA

(e) SGDA     (f) HOGDA-T     (g) HOGDA-HT     (h) HOGDA

Figure 4: The t-SNE visualization of node features learned by SGDA, HOGDA and its two variants on the A→C task (5 classes) with 5% label rate. In all subfigures, the marks ● and × denotes the source domain nodes and target domain nodes, respectively. Fig 4(a-d) illustrate category distributions alignment (Different colors represents different classes). Fig 4(e-h) depict domain alignment (Red: Source domain; Blue: Target domain). *Best viewed in color.*

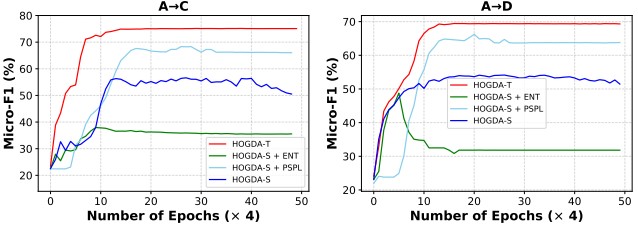

Figure 5: The trend of Micro-F1 during model training.

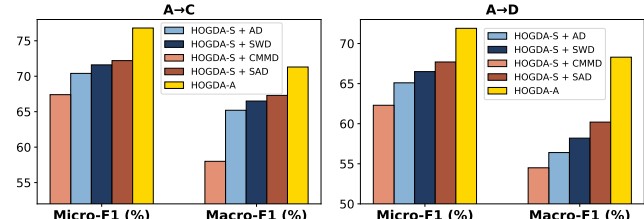

Figure 7: Comparison with different domain alignment strategies on A→C and A→D tasks.

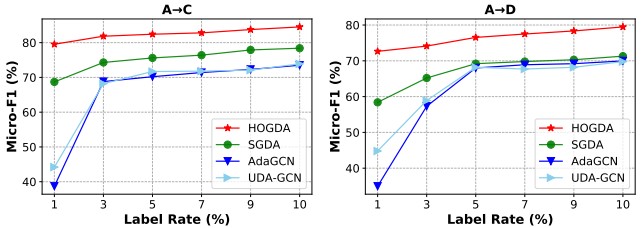

Figure 6: Transfer performance with different label rates.

and evaluate the performance gains brought by these strategies on A→C and A→D tasks, as shown in Figure 7.

We can find that our AWDA strategy significantly outperforms all compared strategies. This is because the compared strategies can only roughly align marginal distributions, while the proposed AWDA can align feature distributions at the class level, which facilitate the model to extract more fine-grained transferable features.

## 5 CONCLUSION

In this paper, we propose a novel model called HOGDA for SGDA. Specially, we introduce a HSIM module to capture high-order structure information in graph in order to better assist the GCN-based feature extractor in learning transferable node features. Additionally, we propose a novel AWDA strategy to encourage the model to pay more attention to easy-to-transfer nodes during adversarial domain alignment, significantly boosting the model's generalization ability on the target domain. More importantly, to address the overfitting issue, a simple but effective TNC strategy is devised to guide the clustering of unlabeled nodes. Comprehensive experiments validate the superiority and stability of our HOGDA on various popular benchmarks.

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
