# OpenReview forum: "HOGDA: Boosting Semi-supervised Graph Domain Adaptation via High-Order Structure-Guided Adaptive Feature Alignmen"
_acmmm.org/ACMMM/2024/Conference — MM2024 Poster_

### Official Review · Reviewer_bSDi · 2024-05-23

**Rating:** 3
**Confidence:** 3

**Summary:**

The paper introduces a novel framework called HOGDA for semi-supervised graph domain adaptation. The framework aims to improve the performance of graph domain adaptation by effectively utilizing high-order structure information and adaptive feature alignment. HOGDA integrates three main strategies: High-order Structure Information Mixing (HSIM), Adaptive Weighted Domain Alignment (AWDA), and Trust-aware Node Clustering (TNC). Extensive experiments on various benchmarks demonstrate that HOGDA outperforms state-of-the-art methods in SGDA tasks.

**Strengths:**

The methodology is thoroughly detailed and the experimental analysis is comprehensive. The overall readability of the paper is high. The combination of HSIM, AWDA, and TNC strategies is novel and effectively addresses the challenges in SGDA. The use of high-order structure information is well-motivated and demonstrates significant performance improvements in capturing complex graph relationships. The paper provides extensive experimental results on multiple benchmark datasets, showing the robustness and superiority of HOGDA over existing methods.

**Limitations:**

- The motivation of the paper is not clear. Many GCN-based methods have already been proposed to fully exploit graph structural information. Can these methods not handle the domain adaptation problem? What specific shortcomings do existing GTL models have in structural mining? I think these issues should be clearly articulated.
- Regarding Section 3.5, the optimization objective Eq. (17) and the loss function Eq. (20) are the same. Is this meaningful? How does this strengthen the robustness of the model?
- How do Eq. (15), (17), and (19) ensure that the obtained $\Phi^*$ possesses the properties described in Lines 543-546 of the manuscript?
- The method lacks complexity analysis and comparison.

**Suitability:**

2

---

### Official Review · Reviewer_Qd4f · 2024-05-25

**Rating:** 5
**Confidence:** 2

**Summary:**

This paper introduces a new framework called HOGDA, which enhances semi-supervised graph domain adaptation by incorporating a High-Order Structure Information Mixing module (HSIM) and an Adaptive Weighted Domain Alignment (AWDA) strategy. The research is novel, the methodology is effective, and the experimental results significantly outperform existing methods. The paper is well-structured and rigorously argued, making it an important contribution to the field of graph domain adaptation.

**Strengths:**

HOGDA introduces a novel approach: it employs the HSIM module to capture high-order structural information, enhancing the transferability of node features; the AWDA strategy focuses on easily transferable nodes, improving the model's generalization ability on the target domain; and the TNC strategy addresses the overfitting issue, guiding the clustering of unlabeled nodes. Through extensive ablation experiments, visualizations, and comparative tests, the effectiveness of HOGDA has been validated.

**Limitations:**

W1: While the overall language is clear and professional, some technical details and mathematical derivations may be overly complex. It is recommended to simplify these parts or provide additional explanations to make the content accessible to a broader audience.
W2: Further exploration could be conducted on the application to other types of graph data and scalability studies on larger datasets.
W3: Although the article discusses the superiority of the method in detail, there is limited discussion on potential applications and social impact. It is suggested to expand this section to highlight the practical significance of the research.
W4: The majority of references cited in the introduction are quite outdated, lacking citations and discussions of the latest research achievements. I recommend that the authors include more recent literature in the introduction to reflect the latest advancements in the field.
W5: The writing in the methods section is not clear enough, overly verbose, and difficult to read. I suggest restructuring the methods section and moving some of the argumentative content to the introduction. Additionally, the value of $\mathcal{J}_0(Z)$ when $k = 1$ is not specified in equation (4). I recommend providing specific values or initialization methods for $\mathcal{J}_0(Z)$.

**Suitability:**

3

---

### Official Review · Reviewer_SiMz · 2024-05-26

**Rating:** 4
**Confidence:** 3

**Summary:**

The paper introduces HOGDA, a framework for semi-supervised graph domain adaptation that addresses the limitations of GCNs-based feature extractors by incorporating high-order graph structure information. HOGDA uses a high-order structure mixing module and an AWDA strategy for dynamic node weight adjustment during adversarial domain adaptation. Additionally, the TNC strategy helps prevent overfitting by guiding unlabeled nodes to form discriminative clusters. Extensive experiments show that HOGDA outperforms state-of-the-art methods in various transfer tasks.

**Strengths:**

1.	HOGDA's integration of high-order structure information and dynamic node weight adjustment represents a significant advancement in graph domain adaptation.
2.	The combination of high-order structure mixing, AWDA strategy, and TNC strategy provides a well-rounded solution addressing multiple challenges in graph transfer learning.
3.	Experimental results show that HOGDA outperforms existing state-of-the-art methods across various transfer tasks, indicating its effectiveness and robustness.

**Limitations:**

1.	The paper does not discuss the scalability of HOGDA in detail, particularly in terms of computational efficiency and applicability to large-scale graphs. A section addressing this aspect with evaluations on large datasets would be beneficial.
2.	The paper does not discuss potential real-world applications of semi-supervised graph domain adaptation or provide examples of real-world datasets where this method could be applied.

**Suitability:**

3

---

### Official Review · Reviewer_eP9P · 2024-05-27

**Rating:** 4
**Confidence:** 3

**Summary:**

This paper focuses on semi-supervised graph domain adaptation, a subfield of graph transfer learning. It addresses the challenge of accurately annotating unlabeled target graph nodes by leveraging transferable features from labelled source nodes. Existing studies often rely on graph convolutional networks (GCNs) for feature extraction, neglecting the collection of complex structure information in graphs. The paper proposes a novel framework called HOGDA that incorporates high-order structure information to enhance transfer learning. Additionally, it introduces the AWDA strategy for fine-grained feature distribution alignment and the TNC strategy to mitigate overfitting. Experimental results demonstrate that HOGDA outperforms state-of-the-art methods in various transfer tasks.

**Strengths:**

1. The paper is well-written and has clear structures and nice figures.

2. The paper is technique-sound with different experiments, and the performance outperforms other models.

**Limitations:**

1. Many notations are presented in the paper. A notation table should be added for better understanding them.

2. Why do you use the encoders with shared parameters? Can you explain more about the motivation?

3. Typos: The Eq. (9) is out of the page scope.

**Suitability:**

2

---

### Meta-Review · Area_Chair_FaYD · 2024-06-27

**Recommendation:** Accept (Poster)
**Confidence:** 5

**Metareview:**

According to all the review comments, rebuttals, discussions and final ratings, the majority of the reviewers gave positive ratings to this paper and the concerns were well addressed. I am happy to recommend to accept this paper. Please carefully revise the final manuscript according to the comments and discussions.